# The Impact of Monetary Benefits in a Pandemic Situation—Navigating Changes in Customer Loyalty through Negative Switching Barriers in the Hotel Industry

**Eunice Minjoo Kang** [1] and **Seul Ki Lee** [2,*]

1   Global Sales, Marriott International, Seoul 03158, Korea; eunice.kang@marriott.com or emk3620@gmail.com
2   Department of Hotel and Tourism Management, Sejong University, Seoul 05006, Korea
*   Correspondence: seulkilee@sejong.ac.kr

**Abstract:** The present study examines the impact of monetary benefits on hotel loyalty programs in a pandemic situation, and the relationship between monetary benefits and multidimensional loyalty aspects. Since 2020, international hotel suppliers have focused on monetary investments and promotions through hotel loyalty programs that have seen extraordinary results in the Korean market. Feedback from the market raises questions as to whether these results are caused by true loyalty or economic sensitivity caused by a recession. Thus, it is necessary to investigate this phenomenon from a relationship perspective. In this research, negative switching barrier facets were given a moderating role with multidimensional loyalty factors consisting of attitudinal loyalty, behavioral loyalty, and composite loyalty. A quantitative method was used, in which consumers in Korea were surveyed via an online questionnaire. In total, 494 responses from consumers were analyzed. Notably, negative switching barriers were found to have a negative moderating effect on the relationship between monetary benefits and attitudinal loyalty. This result implies that although the customer is economically sensitive during a recession, monetary benefits and the market situation can cause fatigue that potentially produces obstacles to long-term relationships with the customer. The results of this study highlight the hotel suppliers involved in monetary competition during a pandemic situation and the need to develop solid long-term relationships through customer loyalty strategies.

**Keywords:** hotel business in COVID-19; hotel loyalty program; monetary benefits; recession; price sensitivity; multidimensional loyalty; relationship marketing; negative switching barrier; sustainability

## 1. Introduction

Since early 2020, the coronavirus (COVID-19) pandemic has affected all aspects of human life. One tremendous impact of the virus was a limitation on travel, which had an unexpected impact on the tourism industry, including, most notably, hotel business. The hotel industry in Korea is no exception to this situation, as demand in this market has dramatically decreased in the country. According to research from the Korea Tourism Organization [1], the total number of travelers in 2019 was 17,502,156, which decreased by 85.6% in 2020. The total number of travelers in 2021 was only 58,397, representing a significant decline of 94.5% compared to 2019. Accordingly, hotel suppliers naturally focused on domestic travelers as domestic travel demands increased due to the limitations on outbound travel. These changes in the market situation then spread to other regions such as Bulgaria and China [2–4].

In such a limited marketplace, hoteliers are responding by developing competitive strategies to attract limited demand through hotel loyalty programs. Competition in the market situation can be categorized as competition between hotel suppliers and online booking agencies. At both ends, the key to competition is for hotel suppliers to promote hotel loyalty programs to attract direct bookings from members. Starting in 2020, international

hotel suppliers such as Hilton®, Marriott®, and Hyatt® have focused on monetary promotions to enhance their hotel loyalty programs [5–7]. For example, Marriott® and Hilton® both launched status challenge promotions, which involved a substantial investment that was considerably different from past initiatives. In addition, Marriott International® launched 'Taste of Platinum', an elite upgrade event for their major corporate customers. This program targets corporate travelers to upgrade from entry-level packages to platinum elite-level packages with just a simple enrolment and only 15 nights of additional required stay. Samsung is one such designated account in Korea [8]. Additionally, at the beginning of 2021, Hilton Honors® launched a fast-track matching status promotion to offer guaranteed gold and diamond levels until 2023 [9].

Similar strategies can be observed in other prominent hotel firms [5–8]. These investments produced quick revenue results from potential loyalty travelers, but the question remains as to whether they will yield long-term customer loyalty.

Uncles, Dowling, and Hammond [10] define a loyalty program as a program offering incentives to encourage customer purchasing and retention. Indeed, many researchers agree that loyalty programs began with 'Green Stamp Frequency', which offered accumulated incentives in line with repeated customer purchasing. In the hospitality field, the 'Advantage Mileage Program' from American Airlines was the first loyalty program, and international hotel suppliers offered membership programs to loyalty customers called 'Hotel Loyalty Programs'. Hence, the first loyalty programs were frequently programs offering monetary benefits for customer retention. To date, many researchers have explored monetary benefits as a motivational factor and examined how it is essential to pursue customer retention by via the pressure of point savings and incentives [11]. Following the link between monetary benefits and customer loyalty, in the Korean market, monetary rewards and cash value play significant roles in hotel loyalty programs [12,13].

However, the majority of recent research has presented different perspectives on service products such as hotel businesses where monetary benefits are not the only factor that affects customer loyalty. In the service industry, the customer expects aspirational or experiential rewards from suppliers of loyalty programs. These rewards are embraced through emotional leverages such as once-in-a-lifetime experiences or lifestyle-themed rewards [14]. From this perspective, customers value emotional benefits such as pleasant service and warm hospitality over a hotel's loyalty program. Indeed, research on relevant trends confirms these perspectives, as differentiated by the characteristics of the industry [14–18].

Although there is competition for monetary benefits and experiential or aspirational rewards among loyalty programs [10], there is no doubt that the current pandemic market situation has significantly affected the hotel industry. According to Ang et al. [19] and Hampson and McGoldrick [20], a concession or crisis environment makes customers more price-sensitive. As outlined in studies on recession and crisis, 'uncertainty' is a significant factor in changes in the market and price sensitivity among customers. With the uncertain atmosphere during the crisis of the pandemic, dynamic environmental changes such as social distancing, lockdown, and limitations on travel have represented new considerations. These limitations and uncertainties have had a significant negative impact on customers who are more price-focused and rational in their choices.

In terms of the pandemic marketplace, the majority of studies have focused on the negative consequences of the economic crisis and its impact on tourism [21,22]. The damage to supply and demand forced the remodeling of hotel loyalty programs to enhance monetary value for overwhelmed customers. Hotel suppliers tried to offer unconventional monetary value promotions through their loyalty programs, which resulted in extraordinary feedback from customers in the pandemic marketplace. However, the question remains as to whether customer loyalty to monetarily beneficial suppliers will last over the long-term and continue to be sustained post-pandemic. Therefore, it is necessary to determine whether this phenomenon is 'true loyalty' or just quick cash investment feedback due to anxiety about the crisis. Still, it is necessary to measure the process of customer loyalty changes if the pressure of switching costs is taken as a negative switching barrier.

Switching costs are conceptualized as the perception of the magnitude of incremental costs required to terminate a relationship and secure an alternative; thus, the dimensions of cash value and monetary benefits are presented through switching costs as price sensitivity that will affect future relationships [23]. From this perspective, switching costs as a major negative switching barrier, and their formation processes, can be observed in all aspects of customer loyalty as a multidimensional aspect. To determine 'true' loyalty, we assess the negative switching barrier formation process by exploring past loyalty-measuring research [24–26]. Exploring this 'true' loyalty is essential for the present study. Additionally, the moderating role of the negative switching barrier is applied to help determine the nexus between attitude, behavior, and long-term loyalty [27].

Since early 2020, with the beginnings of the pandemic marketplace, international hotel suppliers have offered extravagant monetary promotions, which produced quick results in the desire to travel within the Korean market [8,9]. However, it remains questionable whether this outcome is a reflection of customer loyalty or just feedback from incentives or cash value. The purpose of this study is to explore the relationship between monetary benefits and multidimensional loyalty and assess the moderating role of negative switching barrier dimensions on a customer's willingness to exit due to tiredness or trapped feelings. To measure these trapped or unintended feelings and negative attitudinal aspects, it is essential to review multidimensional loyalty and analyze the moderating role of negative switching barriers to measure the interplay between attitudes and behaviors that are powerless against monetary values. The ultimate aim of this research into negative switching barriers and multidimensional loyalty is to explore customers' true loyalty in the pandemic marketplace and facilitate the development of hotel loyalty programs that encourage long-term relationships.

## 2. Conceptual Framework

### 2.1. Hotel Loyalty Programs—Frequency Programs, Monetary Benefits, and Recent Promotions

Hotel loyalty programs began as frequency programs to encourage customer reuse by saving points and mileage [28]. These programs began by offering monetary benefits and expanded to deliver hotel products unlike other products. Recent studies on loyalty programs have focused on dynamics to explain empirically how operational and psychological characteristics support the building of relationships through customer experience [29]. A loyalty program evolves through various stages to build a relationship. The first stages begin with cognitive benefits such as financial benefits. The next stage is the forming of emotional evaluations that support the relationship between the supplier and customer. Cognitive values such as monetary values and emotional evaluations emerge concurrently across the four stages of acquisition, onboarding, expansion, and retention [30,31].

A hotel product's emotional value is different from that of other industrial products. Thus, a customer's expectations of a hotel loyalty program are different from those of loyalty programs in other industries [32]. Such programs offer cognitive financial benefits based on the operational and psychological characteristics offered, which are significant in building a relationship with hotel customers. The greatest benefit of a hotel loyalty program for a hotel company is the creation of a long-term relationship with a loyal customer [33]. This relationship is based on the mutual interests of the hotel business owners and loyal customers.

Although hotel loyalty programs began as frequency programs, similar to airline mileage programs, they were quickly redesigned to include redemption points that could be used with partnering companies such as airlines and car-rental firms. International hotel suppliers continue to refine their loyalty schemes and offer aspirational rewards such as lounge access, free upgrades, and membership events. Some of the world's most famous hotel companies, Hilton Worldwide®, Intercontinental®, and Marriott International®, recently made remarkable changes to their points programs, raising the thresholds for members to receive free-night benefits. L. Zuo, S. Xiong, and H. Iida [34] evaluated the thresholds of the hotel loyalty programs of five international hotel suppliers using gamification theory.

The results suggested that gamification can make an activity more attractive and enjoyable by producing feelings that encourage players to participate more.

An alternative perspective to the relational marketing strategy of hotel loyalty programs is the customer behavior of 'Switching Supplier', which refers to the loss of the supplier's monetary investments and the extra costs involved in retaining customers. These failures mean that it is necessary for the hotel supplier to reconsider monetary investments [35].

### 2.2. Pandemic Situation and Monetary Sensitivity

Customer buying behavior is affected by different factors including cultural and social influences, beliefs, and attitudes [36]. Especially during a recession or social crisis, people tend to be more price-sensitive due to uncertainty. In this context, customers look for cheaper prices and are more concerned about receiving value for their money. Ang et al. [19] and Hampson and McGoldrick [20] further noted that such consumers are more price-conscious and sensitive to sales promotions. The majority of studies have focused on the pandemic as an environmental change, and on its impact on the tourism industry caused by limitations on mobility. Unlike previous recessions, the pandemic crisis threatens customers' physical and mental security [37]. Thus, it is certain that customers' anxiety levels are higher than those pre-pandemic and that price sensitivity has also increased. By the same token, the coronavirus outbreak has not only increased environmental and social anxiety among customers but also impacted customer buying behavior. Additionally, the cancellations of social events forced accommodations and attractions to shut down, having an immediate impact on the value chains of businesses and on people's everyday lives.

The most critical impact on the hospitality value chain system, on the other hand, was the imbalance between supply and demand. Gössling et al. [38] highlighted the decline of the tourism industry, as the virus affected all hospitality value chains. By February 2020, within a month, the perspectives on global tourism changed from over-tourism to non-tourism. Major airlines, including Scandinavian Airlines, Singapore Airlines, and Virgin Airlines, requested aid during the crisis. To date, the contingency situations facing suppliers, including travel restrictions, workforce shortages, increased costs due to hygiene requirements, and oil price increases, have lasted about three years. Additionally, hotel rates are climbing with the return of domestic travelers as the pandemic situation stabilizes [39].

### 2.3. Demand for the Multidimensional Loyalty Scale

Much that has been written about the scale of customer loyalty has focused on customer behavior—i.e., 'consistency' and 'retention'. Retention behavior is essential to measure customer loyalty, but it is not a substitute for true loyalty. Moreover, the problem of a behavioral approach is that repeat purchasing and visiting does not correspond to psychological commitment [32]. Assuming significant psychological commitment, attitudinal loyalty refers to a sense of loyalty, engagement, and preference, although a positive attitude does not mean that a customer will stay with a certain brand when booking a hotel [40]. Pritchard and Howard [41] articulated the need for a composite loyalty approach to measure the predictive power of loyalty, as such an approach would be much more accurate and predictable than scales using behavior or attitude.

Hence, many studies have been reported based on actual proof that 'true loyalty' can be measured with a multidimensional loyalty scale. Moreover, it is believed that multidimensional measurements are much more predictive and accurate for determining customer loyalty [24–27]. Regarding the multidimensional scaling of loyalty aspects, Bowen and Chen [15] highlighted three distinct approaches: behavioral measurement, attitudinal measurement, and composite measurement. These three distinct approaches reflect each scale's advantages and thus enable a review of the nexus between the scaling points.

Behavioral loyalty is measured by a customer's repeated purchasing behavior. The power of behavior offers a direct connection between loyal customers and profitability. According to Bowen and Chen [15], such behavior does not always result from psychologi-

cal commitment to a brand, which means that some repeat purchase cases are processed without loyalty.

Attitudinal loyalty is the decision to reflect emotional and psychological attachment. Attitudinal loyalty relates to the impact of word-of-mouth. Positive word-of-mouth increases the perceived reliability of a hotel and decreases the customer's perceived risk [15]. Oliver [42] suggested recommendation and repurchasing intention alongside premium price as indications of an attitudinal loyalty measurement.

Composite loyalty is sufficient for both behavioral loyalty and attitudinal loyalty and can be defined as repurchasing behavior caused by loyalty engagement [15,43]. A study on customer loyalty usually begins by exploring customer repurchasing and retention behavior. However, from a long-term perspective, this approach is not sufficient for predicting customer loyalty. To compensate for inadequate loyalty predictions, the majority of studies propose using the composite loyalty dimension. According to Pritchard and Howard [41], in service-based industries such as the travel industry, the composite loyalty dimension promotes a better understanding of customer loyalty; the attitudinal dimension is especially useful to distinguish true loyalty. The attitudinal dimension consists of involvement, perceived service differences, and satisfaction scaling. The composite loyalty dimension has predictive power for measuring both behavioral and attitudinal aspects, and represents an essential dimension of the service industry [26,41].

In this study, three distinct loyalty dimensions are used as frames to explore the moderating role of negative switching barriers. In the current pandemic market situation, it is necessary to examine long-term customer loyalty relationships. The inconsistency between loyalty scales is a significant barrier to understanding the status of customer loyalty in the pandemic.

### 2.4. Negative Switching Barriers

Jones et al. [43] define switching barriers as all of the factors that make it more difficult and costly for customers to change service providers. The distinctive switching barriers in the service industry include interpersonal relationships, perceived switching costs, and the attractiveness of competing alternatives, while perceived switching costs and the attractiveness of competitive alternatives are related negative switching barrier composites. The negative switching barrier classification was developed by Hirshman [27], who claimed that a negative switching barrier does not positively affect a customer's repeat purchasing behavior and has a negative moderating impact between customer satisfaction and the loyalty process. A negative switching barrier causes a trapped feeling that forces the customer's behavior, even when the customer is not fully attached to that behavior. Switching costs are a major negative switching barrier [27,44].

Hauser et al. [16] argued that increased costs result in lower price sensitivity in a fixed marketplace, which allows a customer to change their supplier easily, regardless of their satisfaction or loyalty. Additionally, when the marketplace becomes fluid or a major supplier enters the market, the customer feels less empowered due to a lack of alternatives, which also creates a trapped feeling [27,44]. This study adopted a negative switching barrier approach in order to explore the relationship between a hotel loyalty program's monetary benefits and multidimensional loyalty. The negative switching barrier approach is an effective tool for analyzing the current marketplace, which is monetarily invested and an oligopoly of only a few international hotel suppliers such as Marriott International®, Hilton®, Intercontinental®, and Accor®.

The current travel market was affected by the coronavirus pandemic, which limited the mobility of travelers, especially for international travel. Due to the limitations of outbound travel, customers' travel options were limited to domestic travel. However, even domestic travel faced new limitations due to the application of social distancing rules, creating fewer options and available activities than desired.

To relieve travel limitations, countries began to set up 'travel bubbles'. Travel bubbles, also known as 'travel bridges' or 'corona corridors', were developed as a solution

for the current pandemic, as the pandemic is expected to persist for a long period of time. The first travel bubble destination included Estonia, Latvia, and Lithuania. Citizens from these countries were allowed to travel freely within these three countries without quarantine requirements [45].

However, this travel bubble policy involves several processes, including several PCR (Polymerase Chain Reaction) tests to confirm a lack of coronavirus infection. Moreover, the cost is not easily affordable. Additionally, the cost of air tickets to travel bubble destinations is much higher in the post-coronavirus situation. Thus, travel bubbles have caused polarization due to related costs and hurdles, thereby driving the lack of alternatives for customers who desire to travel.

Limitations due to the environmental changes caused by the pandemic have had a significant impact on the lack of alternatives for travelers. Thus, limitations on travel represent a negative switching barrier, as customers feel trapped by their limited travel options, which limit their freedom to decide. The lack of alternatives significantly affects customers in the pandemic situation.

*2.5. Moving Values—Pandemic Security Matters*

Maslow [46] proposed a classification system that reflects the universal needs of society. This system is comprised of four layers in a pyramid, wherein the most basic needs must be met before the individual will strongly desire other activities. The most fundamental demands are the physiological needs sufficient for welfare, such as food, water, rest, and warmth. Next is safety, which refers to a feeling of security in the environment and the needs of safety. The pandemic situation is a recession affecting people's safety. Before the pandemic, people pursued self-actualization, but now they are concerned about their safety from the virus, which is regarded as a basic human instinct: survival. Thus, values are changing to save lives. Money has thus been re-evaluated for its ability to save lives.

In a crisis, customers spend money more sensitively than in normal situations [20]. Additionally, consumers tend to minimize their food expenditures in an economic crisis [47]. Concerns about economic security make customers more price-conscious, as pricing is one of the most significant issues affecting customers during unstable times. Additionally, research on crises shows changes in customers' attitudes to price, and show that customers have an urge to buy through sales activities and seek greater price information [48]. Since 2020, the pandemic crisis has led people to feel insecure and unstable, which may have changed customer attitudes to prices, especially regarding travel, where the limitations on mobility have compelled customers to seek travel alternatives, resulting in a new luxury travel trend called 'hocance' in Korea, which combines the words hotel and vacation, similar to the idea of a 'hotel staycation' [49].

*2.6. Research Hypotheses*

Based on the above literature review and related references, we proposed two hypotheses. The analysis was designed with a two-phase structure. First, the H1 hypothesis was formulated based on the relationship between the monetary benefits of hotel loyalty programs and multidimensional loyalty.

Based on the work of Ang et al. [19] and Kotler and Armstrong [36], the first phase of the hypothesis is composed of the relationships between monetary benefits and multidimensional loyalty. The monetary benefits of hotel loyalty programs are based on international hotel business loyalty program benefits, including Marriott Bonvoy®, IHG rewards club®, Hilton Honors®, and World of Hyatt® [5–7,50].

**Hypothesis 1a (H1a).** *Monetary benefits of hotel loyalty programs have a positive impact on attitudinal loyalty.*

**Hypothesis 1b (H1b).** *Monetary benefits of hotel loyalty programs have a positive impact on behavioural loyalty.*

**Hypothesis 1c (H1c).** *Monetary benefits of hotel loyalty programs have a positive impact on composite loyalty.*

In the secondary phase, we formulated a hypothesis based on negative switching barriers, as applied to the positive impact results of hypothesis H1.

According to Hirshman [27], negative switching barriers emerge from the interplay between customer attitudes and behaviors. Negative switching barriers create a trapped feeling in the current relationship and are largely composed of switching costs and a lack of alternatives. Cost-sensitivity factors were then introduced from research on previous recession cases. Moreover, the pandemic situation has created many limitations for customers, such as limitations on mobility, travel, and potential destinations. These limitations restrict the customer's available options, thus a 'lack of alternatives' is a clear negative switching barrier.

**Hypothesis 2a (H2a).** *Switching costs have a negative moderating effect between monetary benefits and attitudinal loyalty.*

**Hypothesis 2b (H2b).** *Switching costs have a negative moderating effect between monetary benefits and behavioural loyalty.*

**Hypothesis 2c (H2c).** *Switching costs have a negative moderating effect between monetary benefits and composite loyalty.*

**Hypothesis 2d (H2d).** *A lack of alternatives has a negative moderating effect between monetary benefits and attitudinal loyalty.*

**Hypothesis 2e (H2e).** *A lack of alternatives has a negative moderating effect between monetary benefits and behavioural loyalty.*

**Hypothesis 2f (H2f).** *A lack of alternatives has a negative moderating effect between monetary benefits and composite loyalty.*

**3. Methodology**

Based on the above literature review, the following hypotheses were formulated to proceed with quantitative research on Korean customers during the coronavirus pandemic. This quantitative research is based on a self-questionnaire survey carried out between 1 March and 14 April 2021. The total number of participants was 540. Ultimately, the survey results of 494 respondents were selected for further analysis. The purpose of quantitative research in this study is to provide an objective reading of customer's attitudes and behaviors in the pandemic situation based on deductive analysis.

*3.1. Measurement Development*

The items used to measure the monetary benefits of hotel loyalty programs were adopted from previous research on the relationship between monetary benefits and loyalty [12,13]. These items consist of currently offered monetary benefits and cost-savings such as best rate guarantees, food and beverage discount offers, and free breakfasts, as well as frequency savings such as mileage and hotel points [13,51].

The measurement for multidimensional loyalty is adopted from the research of Day [25], Dick and Basu [26], and Baloglu [24]. Statements to scale multidimensional loyalties focus on willingness for reference and retention, and the scaled results are also used to measure composite loyalty.

Negative switching barriers consist of switching costs and a lack of alternatives. Switching costs are measured by the willingness to switch one's supplier due to cost, and

lack of alternatives is measured by the willingness to change one's supplier due to the scale or limitations of the supplier [43,52].

### 3.2. Survey Development and Data Collection

Data were collected from 1 March to 14 April 2021, over a total of 45 days. A survey method using online self-questionnaires was developed due to the circumstances of the pandemic. The survey questions included two segments on monetary benefits and multidimensional loyalty and measured the impacts of negative switching barriers. The respondents were required to have experience with hotels and be adults able to understand the purpose of the survey. The demographic questionnaire included questions on gender, age, salary range, education, and travel behaviors (see Appendix A).

The collected data totaled 540 samples, with a final selection of 494 samples (91.4%). Considering the large sample size of around 500 respondents, the empirical distribution of responses was considered to be normal [53], allowing for the use of parametric tests for data analysis. The data were analyzed using IBM SPSS and AMOS 24.0 software, Seoul, Korea.

## 4. Results

### 4.1. Measurement Model

A confirmatory factor analysis (CFA) was used for the study's proposed model (Figure 1). The results of the CFA based on the data indicated that the proposed model satisfactorily fit the data (F = 4.291, $p$ = 0.002).

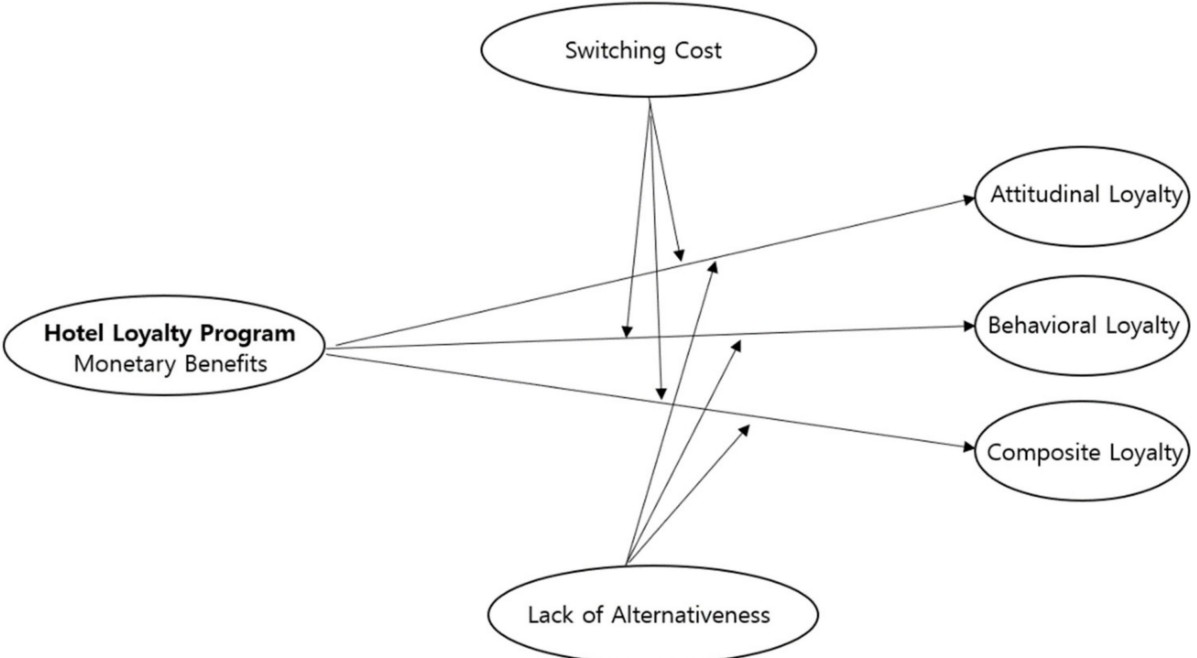

**Figure 1.** Proposed Model. Source: Authors' own elaboration.

### 4.2. Confirmatory Factor Analysis for Monetary Benefit

#### 4.2.1. CFA—1st Step

The first CFA was operated with six measurement items: best rate guarantee, discount benefits of food and beverage in the hotel, bonus point-saving benefits, mileage savings for partnership companies such as airlines and car rentals, guest room upgrade offers, and direct requests to and confirmation of the hotel. The results for the first step indicated that the guest room upgrade offer was not sufficient to be used in factor loading for monetary benefits (0.296). The result of this step is shown in Figure 2.

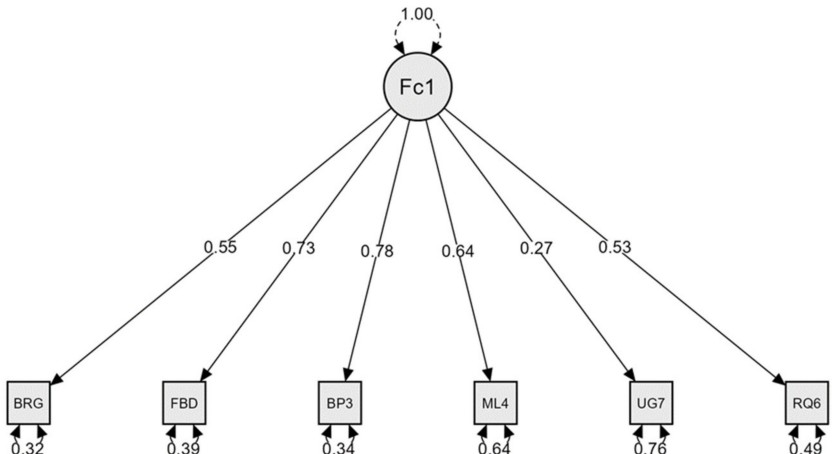

**Figure 2.** 1st step CFA result. Source: Authors' own elaboration.

### 4.2.2. CFA—2nd Step

The guest room upgrade item was eliminated from the first step of the CFA result due to its insufficiency for factor loading. The guest room upgrade item's factor loading result was 0.27, which did not exceed the 0.55 threshold proposed by Falk and Miller for the first step CFA [53]. This result indicates that guest room upgrades are not considered to be a monetary benefit for customers. While developing the second SFA process, the guest room upgrade scale item was eliminated, and the remaining results were sufficient to be used as the monetary benefit factors. The result of this step is shown in Figure 3.

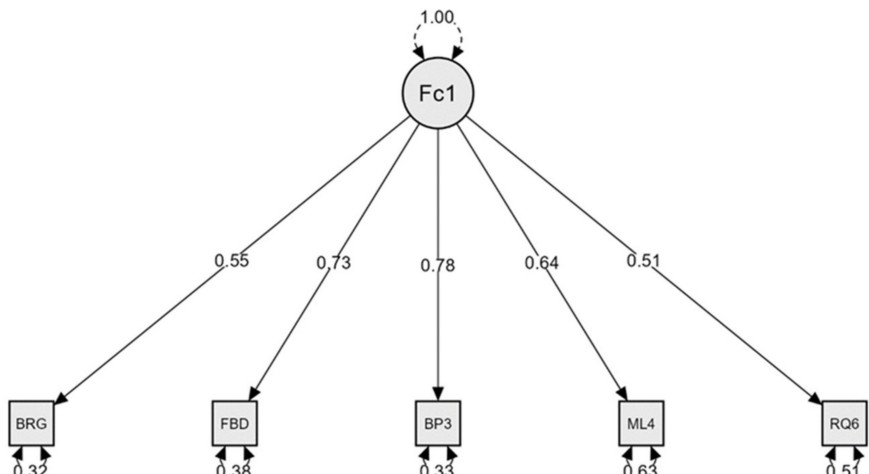

**Figure 3.** 2nd step CFA result. Source: Authors' own elaboration.

The results of the second CFA showed that the measurement model satisfactorily fit the data ($\chi^2 = 18.242$, df = 5, $p < 0.05$, $\chi^2/\mathrm{df} = 3.6484$, RMSEA = 0.073, SRMR = 0.026, GFI = 0.986, IFI = 0.984, NFI = 0.978, and TLI = 0.968, CFI = 0.984). All the factor loadings for indicators from the second CFA were significant at $p < 0.001$. Table 1 shows the AVE (Average Variance Extracted) for each of the indicators, which generally exceeded 0.50. These are based on the cut-off values studied by Fornell and Lacker [54]. Composite reliability was analyzed using the factor loadings and measurement errors for the indicators of each variable, which ranged from 13.161 to 16.310. All items were larger than 1.965 [55]. In the results, all constructs for monetary benefits were measured as having acceptable convergent validity and a satisfactory level of internal consistency (see Table 2).

**Table 1.** Standardized loadings and C.R. and AVE.

| | Indicator | S.E. | *p* | C.R. (AVE) |
|---|---|---|---|---|
| | best rate guarantee | 0.694 | <0.001 | 0.790 (0.433) |
| | discount benefit of food and beverage in hotel | 0.763 | <0.001 | |
| MONETARY BENEFITS | bonus point-saving benefits | 0.806 | <0.001 | |
| | mileage saving for partnership company such as airline, rental car | 0.629 | <0.001 | |
| | direct request to hotel—view, bed type guarantee, higher floor request, non-smoking rooms etc. | 0.586 | <0.001 | |

Source: Authors' own elaboration.

**Table 2.** Results of the measurement model: correlations. Source: Authors' own elaboration.

| Constructs | (1) | (2) | (3) | (4) | (5) | (6) | (7) | (8) | (9) | (10) | (11) |
|---|---|---|---|---|---|---|---|---|---|---|---|
| (1) BRG | — | | | | | | | | | | |
| (2) FBD | 0.547 *** | — | | | | | | | | | |
| (3) BP | 0.573 *** | 0.609 *** | — | | | | | | | | |
| (4) ML | 0.396 *** | 0.446 *** | 0.535 *** | — | | | | | | | |
| (5) RQ | 0.384 *** | 0.477 *** | 0.434 *** | 0.426 *** | — | | | | | | |
| (6) AT | 0.113 * | 0.107 * | 0.059 | 0.056 | 0.111 * | — | | | | | |
| (7) BH | 0.077 | 0.112 * | 0.071 | 0.083 | 0.085 | 0.628 *** | — | | | | |
| (8) CS | 0.103 * | 0.121** | 0.072 | 0.078 | 0.107 * | 0.884 *** | 0.919 *** | — | | | |
| (9) SWC | 0.059 | 0.039 | 0.068 | 0.003 | 0.034 | 0.024 | 0.124 ** | 0.087 | — | | |
| (10) LA | 0.035 | 0.045 | 0.101 * | 0.042 | 0.032 | 0.016 | 0.086 | 0.059 | 0.810 *** | — | |
| (11) XMT | 0.738 *** | 0.806 *** | 0.828 *** | 0.750 *** | 0.704 *** | 0.114 * | 0.112 * | 0.125 ** | 0.052 | 0.068 | — |

* $p < 0.05$, ** $p < 0.01$, *** $p < 0.001$. Source: Authors' own elaboration. BRG = Best Rate Guarantee, FBD = Discount benefits of food and beverage in hotel, BP = Bonus point-saving benefits, ML = Mileage saving for partnership company such as airline, rental car, RQ = Direct request and confirmation to hotel, AT = Attitudinal loyalty, BH = Behavioral loyalty, CS = Composite loyalty, SWC = Switching cost, LA = Lack of alternatives, XMT = Composited monetary benefits.

*4.3. Structure Equation Modeling and Hypothesis Test—Monetary Benefits and Multidimensional Loyalties*

We also performed a structural equation modeling analysis to test the hypothesis of the relationship between monetary benefits and multidimensional loyalties. The overall evaluation of the proposed model and confirmed a satisfactory fit, as shown by the results below (see Table 3).

**Table 3.** Results of the structural model evaluation and hypothesis testing: monetary benefits and multidimensional loyalty.

| Independent Variables | | Dependent Variables | β | t-Value | Status |
|---|---|---|---|---|---|
| H1a monetary benefits | → | attitudinal loyalty | 0.100 | 2.246 * | supported |
| H1b monetary benefits | → | behavioral loyalty | 0.100 | 2.347 * | supported |
| H1c monetary benefits | → | composite loyalty | 0.113 | 2.556 * | supported |

* $p < 0.05$. Source: Authors' own elaboration.

(1)  H1a: Monetary benefits of hotel loyalty programs have a positive impact on attitudinal loyalty—supported.

Overall confirmed: $\chi^2$ = 22.746, df = 9, $p < 0.05$, $\chi^2/df$ = 2.527, RMSEA = 0.056, NFI = 0.973, IFI = 0.984, CFI = 0.983, TLI = 0.972.

(2)  H1b: Monetary benefits of hotel loyalty programs have a positive impact on behavioral loyalty—supported.

Overall confirmed: $\chi^2$ = 19.853, df = 9, $p < 0.05$, $\chi^2/$df = 2.205, RMSEA = 0.049, NFI = 0.976, IFI = 0.987, CFI = 0.987, TLI = 0.978.

(3)  H1c: Monetary benefits of hotel loyalty programs have a positive impact on composite loyalty—supported.

Overall confirmed: $\chi^2$ = 21.010, df = 9, $p < 0.05$, $\chi^2/$df = 2.334, RMSEA = 0.052, NFI = 0.975, IFI = 0.986, CFI = 0.986, TLI = 0.976.

The constructs of monetary benefits consisted of best rate guarantees, discount benefits of food and beverages in hotels, bonus point-saving benefits, mileage savings for partnership companies, and direct requests to and confirmation of the hotel. All three hypotheses were verified at $p < 0.05$.

*4.4. Moderating Effects of Negative Switching Barriers*

4.4.1. The Moderating Role of Switching Costs between Monetary Benefits and Multidimensional Loyalty

The negative switching barrier dimensions consisted of switching costs and a lack of switching alternatives. Based on the supported results of previous analyses, multiple group analyses for cross-group equality constraints were employed to examine the moderating role of the switching cost item. As a negative switching barrier, the switching cost item was found to moderate the relationship between monetary benefits and attitudinal loyalty: $\Delta\chi^2[1]$ = 10.126 and $p < 0.005$. The results for composite loyalty were $\Delta\chi^2[1]$ = 9.359 and $p < 0.005$. However, for behavioral loyalty, the moderating role of switching cost did not support the hypothesis: $\Delta\chi^2[1]$ = 5.553 (see Table 4).

**Table 4.** Results of the structural model evaluation and hypothesis testing: moderating role of switching cost in relationship of monetary benefits and multidimensional loyalty.

| Linkages w: Switching Cost | Baseline Model (Freely Estimated) | Nested Model (Equally Constraint) |
|---|---|---|
| H2a Monetary benefits → attitudinal loyalty | $\chi^2(35)$ = 46.510 | $\chi^2(36)$ = 53.266 |
| H2b Monetary benefits → behavioral loyalty | $\chi^2(35)$ = 52.137 | $\chi^2(36)$ = 57.690 |
| H2c Monetary benefits → composite loyalty | $\chi^2(35)$ = 49.112 | $\chi^2(36)$ = 58.471 |

(1) H2a: Switching cost has a negative moderating effect between monetary benefits and attitudinal loyalty—supported $\Delta\chi^2[1]$ = 10.126, $p < 0.005$. (2) H2b: Switching cost has a negative moderating effect between monetary benefits and behavioral loyalty—rejected $\Delta\chi^2[1]$ = 5.553. (3) H2c: Switching cost has a negative moderating effect between monetary benefits and composite loyalty—supported $\Delta\chi^2[1]$ = 9.359, $p < 0.005$. Source: Authors' own elaboration.

4.4.2. The Moderating Role of a Lack of Alternatives between Monetary Benefits and Multidimensional Loyalty

As a major negative switching barrier, a lack of alternatives was assumed to be significant, as the market situation is becoming more option-limited than it was in the pre-pandemic situation. Thus, we assumed that this component would have a moderating role on the relationship between monetary benefits and multidimensional loyalty. Based on the results of previous analyses, multiple groups of analyses for cross-group equality constraints were employed to examine the moderating role of a lack of alternatives. The lack of alternatives item as a negative switching barrier was found to moderate the relationship between monetary benefits and attitudinal loyalty: $\Delta\chi^2[1]$ = 4.481, $p < 0.025$. However, for behavioral loyalty and composite loyalty, the results for this item did not support the hypothesis: $\Delta\chi^2[1]$ = 0.325, $\Delta\chi^2[1]$ = 2.083 (see Table 5 and Figure 4).

**Table 5.** Results of the structural model evaluation and hypothesis testing: moderating role of lack of alternatives in the relationship of monetary benefits and multidimensional loyalty. Source: Authors' own elaboration.

| Linkages w: Lack of Alternatives | Baseline Model (Freely Estimated) | Nested Model (Equality Constrained) |
|---|---|---|
| H2d Monetary benefits → attitudinal loyalty | $\chi^2(35) = 43.140$ | $\chi^2(36) = 50.991$ |
| H2e Monetary benefits → behavioral loyalty | $\chi^2(35) = 49.927$ | $\chi^2(36) = 50.252$ |
| H2f Monetary benefits → composite loyalty | $\chi^2(35) = 48.265$ | $\chi^2(36) = 50.348$ |

(1) H2d: Lack of alternatives has a negative moderating effect between monetary benefits and attitudinal loyalty—supported $\Delta\chi^2[1] = 4.481$, $p < 0.025$. (2) H2e: Lack of alternatives has a negative moderating effect between monetary benefits and behavioral loyalty—rejected $\Delta\chi^2[1] = 0.325$. (3) H2f: Lack of alternatives has a negative moderating effect between monetary benefits and composite loyalty—rejected $\Delta\chi^2[1] = 2.083$. Source: Authors' own elaboration.

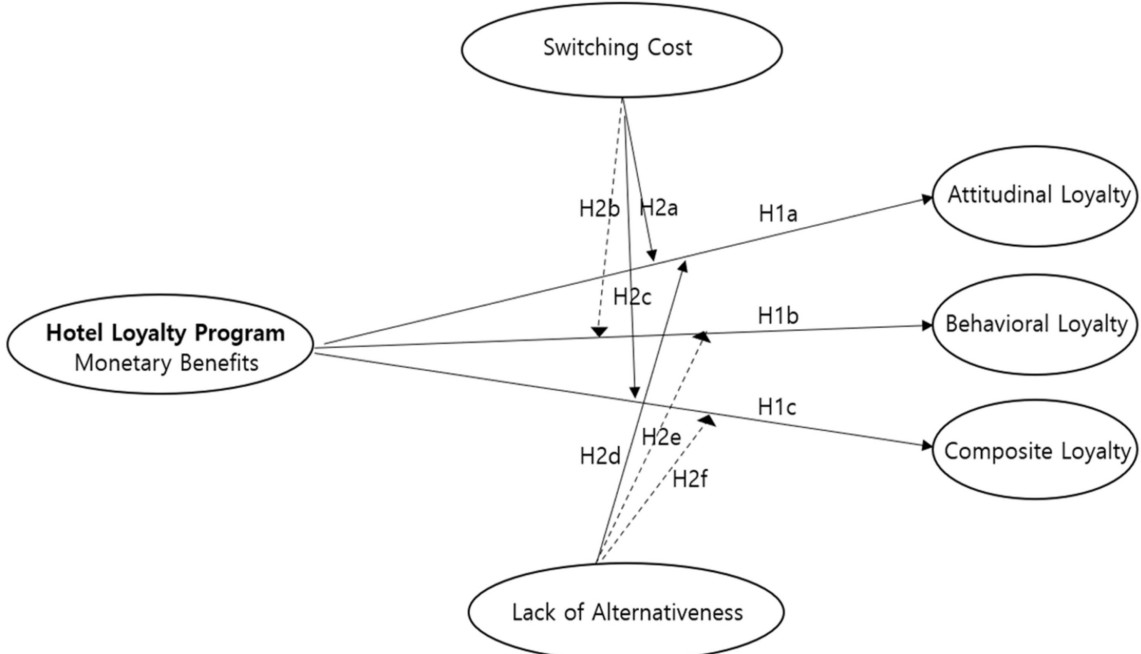

**Figure 4.** Result of hypothesis testing. Source: Authors' own elaboration.

## 5. Discussion and Implications

### 5.1. Discussion

The results of the study were acquired in two phases. The first was a regression analysis of the relationship between the monetary benefits of hotel loyalty programs and multidimensional loyalty. Loyalty aspects were reviewed using the three scales of attitudinal loyalty, behavioral loyalty, and composite loyalty. In this process, monetary benefits were found to influence all aspects of multidimensional loyalty. This result shows that the pandemic-era marketplace in Korea is increasingly sensitive to price, with a trend similar to the way in which a crisis environment increases customer sensitivity to price [20,47,56]. Additionally, this change in behavior during the pandemic suggests that the virus situation has forced security to become more greatly valued [46] while also increasing customer anxiety and dissatisfaction. As a pandemic poses a risk to one's life, there is a much higher stress level during a pandemic situation than during other crises such as economic downturns or political tension.

Another aspect of monetary benefits is extrinsic rewards, which affect how calculative customers are [55]. Since 2020, international hotel businesses have offered many monetary promotions and fast-track upgrades. These changes are linked to Korean travelers' anxieties and their desire to travel [57].

Additionally, hotel products classify 'precommitment and self-control' as a luxury good. According to Thaler [58], the customer feels guilty when purchasing luxury goods such as a hotel product. To ease this feeling, customers buy hotel products and services in advance and apply a calculative process. Since 2020, there have been many voucher promotions offering flexibility to book with any schedule during certain periods due to the pandemic situation. These promotions were successful, as customers bought cheaper vouchers more frequently than during the pre-pandemic period because of their desire to travel.

On the other hand, the results of the moderating role of negative switching barriers indicated a reversal of the current relationship between monetary benefits and multidimensional loyalty. Switching cost, as a major negative switching barrier, has a negative moderating role for attitudinal loyalty and composite loyalty but does not affect behavioral loyalty. Apparently, customers remain firmly loyal to their current suppliers, but their attitudes and general judgement are negative toward monetary stimulation and calculative reactions. This result is consistent with research defining the formation process of negative switching barriers. The inconsistency between attitude and behavior creates feelings of being forced or trapped [27,44,59].

The secondary results for the moderating role of the lack of alternatives between monetary values and multidimension loyalty were also limited but followed the trends observed in the moderating role of switching costs. The moderating role of a lack of alternatives was found to have a negative moderating effect between monetary benefits and attitudinal loyalty, as well as composite loyalty, but was negligible in significance. Besides these results, a lack of alternatives did not have a moderating role between monetary benefits and behavioral loyalty. The lack of alternatives dimension has significance in the pandemic marketplace, under which there are new limitations to mobility and selection. In our research, the questionnaires focused on the limitations in selection, including the supplier's size and travel options. A survey of Booking.com [57] found that the desire to travel is much higher, but customers prefer familiar destinations due to risk management. This also indicates that customers experience feelings of being trapped even if they also have an increased wish to travel. Currently, customers experience negative feelings due to a lack of alternatives; however, this result this does not affect the item's moderating role on behavioral and composite loyalty, based on our understanding of the COVID-19 crisis.

According to Vaszquez and Foxall [59], the role of negative switching barriers is possibly amplified in the service industry. In the service industry, there is a relationship and experience implied between the supplier and customer. Even if the service is dissatisfactory, the customer can be retained due to their relationship and expectations based on previous experiences. Indeed, customers have different expectations of the service industry, and previous research has shown that the relationship value between service suppliers and customers is key.

### 5.2. Managerial Implications

First, this work is differentiated from previous research in that the present study was designed to review the three dimensions of customer loyalty and measure loyalty by moderating the impacts of negative switching barriers under the current pandemic situation. Additionally, during the pandemic, perspectives on customer–supplier relationships determined the significance of the customer's psychological impact and attitude changes. The results indicate that the customer's state of mind is not comparable to that during conventional crises such as economic recessions. The pressure of security and uncertainty drives feelings of fatigue in customers engaging in price competition, so it is necessary to review customer loyalty from a relationship perspective, and not only from the perspective of loyalty.

The results of this study highlight hotel loyalty programs as a relationship marketing tool from a long-term perspective. Developing a customer relationship management (CRM) perspective is necessary in order to manage long-term relationships developed through

hotel loyalty programs. In the digital environment, offering information to customers is a significant factor, and an information service strategy based on CRM could be utilized. According to Saura et al. [60], internal control factors and external control factors are significant in a CRM strategy. Internal factors can represent the service capacity of hotel loyalty programs, which is a strategy using an omni-channel service that influences long-term relationships with hotel customers in a competitive marketing environment (which is an external control factor). To deliver the internal control factors to hotel customers, strategic CRM materialization is necessary for hotel loyalty programs.

The relationship between monetary benefits and customer loyalty in multi-dimensional scales was apparently positive. However, along with the moderating role of negative switching barriers, this relationship causes negative feelings among customers in the long-term. One aspect of negative switching barriers is the degree to which customers experience a sense of being locked into a relationship with a supplier based on various costs associated with exiting the relationship. According to the results, this feeling can be changed by moderating the role of negative switching barriers. The results indicated that customers experience negative feelings, rather than satisfaction, in their existing relationships, hinting at the possibility of terminating the relationship. Thus, it is necessary for hotel suppliers to focus on enhancing their relational benefits and service attributes, rather than monetary offers. Experiential events and promotions also encourage customer loyalty in hotel loyalty programs.

## 6. Conclusions

This research highlights the need for sustainability risk management (SRM) in the marketplace. This measure is related not only to ecological impacts but also to economic impacts on the sustainability of the hotel market between suppliers, customers, and hotel owners. Economic sustainability is one of the major pillars of sustainability, which also consists of human sustainability, social sustainability, and environmental sustainability. The point of economic sustainability is to build a sustainable development marketplace for all stakeholders and the environment. If suppliers and customers focus on price and cost competition, this will have a negative impact and become a hindrance to the marketplace. Additionally, economic sustainability requires consideration of long-term relationships. Aiming for instant results through monetary competition will negatively impact the growth and wellbeing of the marketplace. From this perspective, it is recommended to encourage the implementation of sustainability risk management among hotel owners to overcome this critical situation from a long-term perspective. In a pandemic situation, this sustainability perspective is also key to enhancing the relationship between the supplier and loyal customers. The dialogue from this research shows that stressful environments have an ambivalent impact on customers, who experience desire and nervosity. Therefore, investments in cleanliness and ecological improvements, for both the environment and the marketplace, will help relieve anxiety in the crisis market.

There are also limitations arising from the pandemic situation and the oligopoly of the present marketplace. The oligopoly of the hotel industry progressed rapidly before the pandemic. The scale of this industry's economy was an advantage to a few major hotel suppliers, but COVID-19 has changed the market situation. Based on the results, customers have been affected by a lack of alternatives and/or a lack of opportunities. Hence, customers have become more sensitive to feeling powerless and having limited options. Based on our research, composite-operation hotel loyalty programs are possibly not positive for customers since they may produce pressure and stress in this limited marketplace.

*Limitations and Future Studies*

This study used a quantitative approach based on a self-directed survey. Since the pandemic, there have been limitations in conducting surveys on hotel loyalty programs. Out of the 494 total respondents, 33.4% of respondents were loyalty program members, and 66.4% were non-members who had experience with booking and staying in hotels.

The findings show that it is necessary to develop more indicators to measure the monetary benefits of hotel loyalty programs. Recent trends in hotel loyalty programs include point savings and redemptions in hotel food and beverage outlets. Due to the limitations of travel and related risks, customers look for special atmospheres in hotels, and hotel suppliers provide motivation via promotions through hotel loyalty programs. Adding hotel food and beverage outlets as dimensions would provide an opportunity to better understand changes in customer loyalty and thus explore more specific applications of scaled data based on overall market demand.

The present survey took place from March to April 2021, during the COVID-19 pandemic. This unconventional situation affected the research results, which means that this subject remains open for further analysis in a post-pandemic context. The present results indicated predictable changes in customer loyalty, so future evidence will better support empirical research results on this topic. Additionally, this study surveyed Korean customers, so future studies should examine diverse market situations to support a broader understanding of changes in global trends during the pandemic.

**Author Contributions:** Conceptualization, E.M.K. and S.K.L.; methodology, E.M.K.; formal analysis, E.M.K.; writing—original draft preparation, E.M.K.; writing—review and editing, S.K.L.; supervision, S.K.L. All authors have read and agreed to the published version of the manuscript.

**Funding:** This research received no external funding.

**Informed Consent Statement:** Informed consent was obtained from all subjects involved in the study.

**Conflicts of Interest:** The authors declare no conflict of interest.

## Appendix A

Survey for a doctoral dissertation

Eunice Minjoo Kang Ph.D.

Dr. Seul Ki Lee, professor/department of hotel and tourism management, Sejong University

This survey was created for the study of 'the effect of hotel loyalty program's selection attributes on customer loyalty' and moderating impact of switching barriers.

Through this study, we analyzed the motivating factors and constraint factors, which are the selection attributes of the hotel loyalty program, and based on this, we established the future direction of the hotel loyalty program and analyzed the moderating effect of the switching barrier to determine the external factors of the hotel loyalty program. By analyzing at the same time, we expect to be able to provide strategic proposals for the program.

Your responses will be used as very important data in this study, so please answer every single question.

All information related to this survey will be used for academic purposes only for research purposes, and all your personal information will be treated anonymously and will be used for academic purposes only. Therefore, your responses to the questionnaire are protected by the statistical act of law 33 (Protection of Confidentiality).

Thank you very much for responding to the survey.

1. **The following are demographic questions. Please read each question carefully and mark (√) the appropriate number correctly**.

| | Questions | Answer | | | | |
|---|---|---|---|---|---|---|
| 1. | gender | ① male | | ② female | | |
| 2. | age | 20's | 30's | 40's | 50's | 60's |
| 3. | marriage status | single | | married | | no answer |
| 4. | education | high school graduated | college graduated | university graduated | | master degree |

2. **The following questions are about the usage behavior of the hotel loyalty program. Please read each question carefully and mark (√) the appropriate number correctly**.

The Hotel Loyalty Program is a membership program offered by international hotel group companies such as Marriott Bonvoy, IHG Rewards Club, and Hilton Honors.

(1) Are you a member of a hotel loyalty program?
① Yes       ② No

(2) Are you using the Hotel Loyalty Program?
① Yes       ② No

(3) If you are using a hotel loyalty program, please indicate which hotel loyalty program you mainly use.

| Hotel Brand | Hotel Loyalty Program | Duration as a Member of Program |
| --- | --- | --- |
| Marriott International | Marriott Bonvoy | |
| Hyatt corporation | World of Hyatt | |
| Hilton hotel & resort group | Hhonors | |
| Intercontinental hotel group | IHG Rewards | |
| Accor hotels and group | ALL loyalty program | |
| etc. | | |

(4) Please enter the year when you joined the hotel loyalty program.
(5) Please select the purpose for which you mainly use the hotel loyalty program.

① business ② leisure ③ F&B ④ etc.

3. **The following is the description of the monetary benefits of the hotel loyalty program.**

| Monetary Benefits of The Hotel Loyalty Program |
| --- |
| Best Rate Guarantee benefits for members is hotel using an auditing system for loyalty members that guarantees the lowest price among the prices exposed on the web, Provides mileage that can be used at other partner companies during the stay, bonus points benefit when making a reservation using the loyalty program, and free resort fee. Benefits include monetary rewards and rapid upgrades for elite loyalty of the members such as status challenges or taste of elite member promotion. |

(1) The following are detailed questions for evaluation of the loyalty program's monetary benefits. Please mark (√) the appropriate number correctly.

| Dimensions | 1.Surely Not Important | 2. Not Important | 3.Average | 4.Important | 5.Very Important |
| --- | --- | --- | --- | --- | --- |
| 1. Best Rate Guarantee for members | | | | | |
| 2. F&B discount | | | | | |
| 3. Offer bonus point | | | | | |
| 4. Milage offer available redemption to airlines and rent a car | | | | | |
| 5. Direct service request and confirmation from the hotel for members | | | | | |
| 6. Upgrade for loyalty members | | | | | |

4. **The following question is about your loyalty scaling for the hotel loyalty program**.

(1) I will highly recommend it to other potential customers when I have identified the monetary benefits of the hotel loyalty program.

1. Surely not recommended. 2. not recommended 3. can't decide it 4. recommended. 5. highly recommended.

(2)    I will keep using the loyalty program when I understand the monetary benefits of the hotel loyalty program.

1. Surely not use it 2. not use it 3. can't decide it 4. will use it 5. Surely will use it.

5.    **The following is a questionnaire on the overall effect of switching barriers on the loyalty of hotel loyalty programs.**

(1)    I understand that if you change from the current channel to another channel and use it, there is a big loss in money and time.

1. Surely not agree 2. not agree 3. moderate 4. agree 5. surely agree.

(2)    I believe that the benefits of the current channel cannot be provided by other channels.

1. Surely not agree 2. not agree 3. moderate 4. agree 5. surely agree.

(3)    I will not move to another channel because the number of available hotels provided by the current channel is large and there are various brands.

1. Surely not agree 2. not agree 3. moderate 4. agree 5. surely agree. Sources own elaboration.

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
