# Peer review of "The Impact of Monetary Benefits in a Pandemic Situation—Navigating Changes in Customer Loyalty through Negative Switching Barriers in the Hotel Industry"

_sustainability, doi:10.3390/su14138079_

Round 1
Reviewer 1 Report
Dear authors:
Manuscript is quite interesting but there are some parts that must be improved:
- English must be improved
- Title is too long and descriptive
- Method and analysis must be improved in order to narrow down hypotheses and adequately explain what exactly is the authors' contribution to social and marketing science in this research
- The arguments and discussion of findings must be also improved
- It is essential to draw some conclusions from the work, apart from the discussion section
Author Response
Dear Reviewer,
Thank you for your insightful comments on our manuscript. We tried our best to address them as follows.
Manuscript is quite interesting but there are some parts that must be improved:
1. English must be improved
=> We thoroughly edited the manuscript for grammar and expression.
2. Title is too long and descriptive
=> We amended title more impact and simple.
3. Method and analysis must be improved in order to narrow down hypotheses and adequately explain what exactly is the authors' contribution to social and marketing science in this research
=> Method and analysis are updated for proper explanations for hypotheses.
4. The arguments and discussion of findings must be also improved
=> We reviewed discussion and conclusión and add updates from customer management relationship strategy references,
5. It is essential to draw some conclusions from the work, apart from the discussion section
=> We reviewed discussion and conclusión and add updates from customer management relationship strategy references.
Once again, thank you very much for your input.
Reviewer 2 Report
Article: The impact of monetary benefits in pandemic situation - How do negative switching barriers affect relationship between monetary benefit and loyalty in the hotel industry during the pandemic?
Journal: sustainability-1676450
Reviewer: Prof. Dr. Juan Carlos Meléndez
First Review
The authors of this article have presented an interesting research about the impact of monetary benefit in pandemic situation toward hotel loyalty program.
Please, find below some comments and suggestion to be considered by the authors:
1.- All the authors referenced in the last section of the article must be listed in the same order as they appear in the original document. Please star in line 35 with ref. [1] and continue sequentially. It is very difficult to follow the references referenced in an unsorted list.
2.- All the references included in the last part are not following the rules defined in the Style Guide for MDPI Journals, please follow the main rule: “Title of the article. Journal Abbreviation Year, Volume”. This must be amended.
3.- In line 39 is mentioned Korea market, however the ref. 31 is referred to a Bulgaria case. This is not coherent.
4.- line 46: Are the Hilton, Marriott, Hyatt names protected or registered brands? If yes, please, add “®”. Please, apply this concept to all the manuscript.
5.- The introduction section includes several references to loyalties program without any order or key parameters. I strongly recommend to identified these key parameters and shown them in a summary table.
6.- I miss any academicals reference to the theory of Multidimensional Brand Loyalty of Sneth and Park. They are authors who are cited many times.
7.- The section 2.2 does not clearly demonstrate the negative effect of pandemic in the monetary sensitive. It means that I miss some bar diagrams with figures to show this impact.
8.- The authors should clearly indicate if tables, diagrams and figures are “source own elaboration” or not. Please review all figures and tables. And also, the tables are not following the template of the Journal. Please review the guideline information for authors.
9.- I cannot check if the survey used in this research are properly made to focus on the main hypotheses formulated in the article. In another word, I recommend to include in an annex the questionnaire to confirm that the attitudinal loyalty, behavioral loyalty, and composite loyalty are properly analyzed.
10.- Author Contributions, Funding and Conflicts of Interest sections are missed. Please, include them.
In general, after reading the manuscript, I recommend the authors review the content in order to amend the following issues:
- Academicals references properly sorted.
- To organize properly the literature review including a summary table with the key topics.
- To demonstrate that the survey used as input in this research are built to measure the relationship between the tree loyalties and the economics behavior.

Author Response
Dear Reviewer,
Thank you for your insightful comments on our manuscript. We tried our best to address them as follows.
The authors of this article have presented an interesting research about the impact of monetary benefit in pandemic situation toward hotel loyalty program.
1. All the authors referenced in the last section of the article must be listed in the same order as they appear in the original document. Please star in line 35 with ref. [1] and continue sequentially. It is very difficult to follow the references referenced in an unsorted list.
=> We amended terms for references,
2. All the references included in the last part are not following the rules defined in the Style Guide for MDPI Journals, please follow the main rule: “Title of the article. Journal Abbreviation Year, Volume”. This must be amended.
=> We revised the references from style guide.
3. In line 39 is mentioned Korea market, however the ref. 31 is referred to a Bulgaria case. This is not coherent. => Bulgaria case is similar travel behavior change such as Korea case since limitation of outbound travel is global travel environment due to pandemic, this reference explains that limitation of outbound travel motivates inbound travel. We see it is necessary to change description clear, it is now updated.
4. line 46: Are the Hilton, Marriott, Hyatt names protected or registered brands? If yes, please, add “®”. Please, apply this concept to all the manuscript.
=> It is understood that the hotel brands as general, common names, so it is already protected and thus there is no necessary to mark with ®.
5. The introduction section includes several references to loyalties program without any order or key parameters. I strongly recommend to identified these key parameters and shown them in a summary table.
=> We reviewed and updated the parameters.
6. I miss any academicals reference to the theory of Multidimensional Brand Loyalty of Sneth and Park. They are authors who are cited many times.
=> We reviewed the articles on multidimensional brand loyalty by Sneth and Park, it is added for multidimensional loyalty reference.
7. The section 2.2 does not clearly demonstrate the negative effect of pandemic in the monetary sensitive. It means that I miss some bar diagrams with figures to show this impact.
=> We updated the description for clear explanation.
8. The authors should clearly indicate if tables, diagrams and figures are “source own elaboration” or not. Please review all figures and tables. And also, the tables are not following the template of the Journal. Please review the guideline information for authors.
=> We updated templates from guideline.
9. I cannot check if the survey used in this research are properly made to focus on the main hypotheses formulated in the article. In another word, I recommend to include in an annex the questionnaire to confirm that the attitudinal loyalty, behavioral loyalty, and composite loyalty are properly analyzed.
=> The references from the loyalty facet are updated and survey is based on previous studies and references, we updated conceptual framework part.
10.- Author Contributions, Funding and Conflicts of Interest sections are missed. Please, include them.
=> They are now included.
In general, after reading the manuscript, I recommend the authors review the content in order to amend the following issues:
Academicals references properly sorted.
=> The recommended references are updated.
To organize properly the literature review including a summary table with the key topics.
=> Literature reviews are updated, and related references are added accordingly.
To demonstrate that the survey used as input in this research are built to measure the relationship between the tree loyalties and the economics behavior.
=> The dimensions are consisted of empirical researches and the references are added to support.
Once again, thank you very much for your input.
Reviewer 3 Report
I hope you're well. I appreciate your scientific effort to produce this manuscript. Your paper has scientific value and interesting.
This manuscript identified the impact of monetary benefit in pandemic situation toward hotel 10 loyalty program, its monetary benefits to multidimensional loyalty dimensions.
Overall, the research work does not adequately review the literature, so we propose to improve both the quality and quantity of references and, above all, to clarify to the reader what is necessary to refresh strategy of relationship marketing perspective.
First, the authors need to better describe what the dimensions of loyalty are. They should also justify why they have chosen confirmatory factor analysis (CFA) and SEM as a method of statistical analysis. Above all, they can justify different fields of application of SEM, such as tourism:
Palos-Sanchez, P., Saura, J. R., Velicia-Martin, F., & Cepeda-Carrion, G. (2021). A business model adoption based on tourism innovation: Applying a gratification theory to mobile applications. European Research on Management and Business Economics, 27(2), 100149.
Loyalty in restaurants:
Hernandez-Rojas, R. D., Folgado-Fernandez, J. A., & Palos-Sanchez, P. R. (2021). Influence of the restaurant brand and gastronomy on tourist loyalty. A study in Córdoba (Spain). International Journal of Gastronomy and Food Science, 23, 100305.
Reyes-Menendez, A., Palos-Sanchez, P. R., Saura, J. R., & Martin-Velicia, F. (2018). Understanding the influence of wireless communications and Wi-Fi access on customer loyalty: a behavioral model system. Wireless Communications and Mobile Computing, 2018.
Loyalty in sports:
Martín, F. V., Toledo, L. D., & Palos-Sanchez, P. (2020). How deep is your love? Brand love analysis applied to football teams. International Journal of Sports Marketing and Sponsorship.
Secondly, this journal deals with issues related to sustainability, the authors can reference this work published in Sustainability Journal and see examples of topics related to hotels and sustainability. It is important that the manuscript justifies how it addresses this issue.
Ríos-Martín, M. Á., Folgado-Fernández, J. A., Palos-Sanchez, P. R., & Castejon-Jimenez, P. (2019). The impact of the environmental quality of online feedback and satisfaction when exploring the critical factors for luxury hotels. Sustainability, 12(1), 299.
Some suggestions are as follows:
Please use different terms in the "Title" and the "Keywords".
Please use the full term when use for first time an abbreviation.
The abstract should state briefly the purpose of the research, the principal results and major conclusions. An abstract is often presented separately from the article, so it must be able to stand alone.
The Introduction should end with a paragraph devoted to explaining the structure of the manuscript to the reader.
Authors should justify why they have chosen confirmatory factor analysis (CFA), based on previous work. Some are provided above. They should also indicate if there are items that were eliminated, giving reasons why.
Authors should explain what type of sampling they used, giving a rationale as to why it was chosen.
What software was used to do confirmatory factor analysis (CFA) and SEM. In the manuscript they state: "The data were analysed by SPSS 24.0 and ANOVA", but ANOVA is not a software, but a statistical technique. Please clarify what ANOVA was used for.
It is suggested to compare the results of the present research with some similar studies which is done before.
What role can CRM systems play to refresh strategy of relationship marketing perspective and
in achieving the objectives this manuscript concludes on?
Saura, J. R., Palos-Sanchez, P., & Blanco-González, A. (2019). The importance of information service offerings of collaborative CRMs on decision-making in B2B marketing. Journal of Business & Industrial Marketing.
It is suggested to organize Conclusion section much better. This section should present in one 250-300 words paragraph.
Author Response
Dear Reviewer,
Thank you for your insightful comments on our manuscript. We tried our best to address them as follows.
I hope you're well. I appreciate your scientific effort to produce this manuscript. Your paper has scientific value and interesting.
This manuscript identified the impact of monetary benefit in pandemic situation toward hotel 10 loyalty program, its monetary benefits to multidimensional loyalty dimensions.
Overall, the research work does not adequately review the literature, so we propose to improve both the quality and quantity of references and, above all, to clarify to the reader what is necessary to refresh strategy of relationship marketing perspective.
First, the authors need to better describe what the dimensions of loyalty are. They should also justify why they have chosen confirmatory factor analysis (CFA) and SEM as a method of statistical analysis. Above all, they can justify different fields of application of SEM, such as tourism:
Palos-Sanchez, P., Saura, J. R., Velicia-Martin, F., & Cepeda-Carrion, G. (2021). A business model adoption based on tourism innovation: Applying a gratification theory to mobile applications. European Research on Management and Business Economics, 27(2), 100149.
Loyalty in restaurants:
Hernandez-Rojas, R. D., Folgado-Fernandez, J. A., & Palos-Sanchez, P. R. (2021). Influence of the restaurant brand and gastronomy on tourist loyalty. A study in Córdoba (Spain). International Journal of Gastronomy and Food Science, 23, 100305.
Reyes-Menendez, A., Palos-Sanchez, P. R., Saura, J. R., & Martin-Velicia, F. (2018). Understanding the influence of wireless communications and Wi-Fi access on customer loyalty: a behavioral model system. Wireless Communications and Mobile Computing, 2018.
Loyalty in sports:
Martín, F. V., Toledo, L. D., & Palos-Sanchez, P. (2020). How deep is your love? Brand love analysis applied to football teams. International Journal of Sports Marketing and Sponsorship.
Secondly, this journal deals with issues related to sustainability, the authors can reference this work published in Sustainability Journal and see examples of topics related to hotels and sustainability. It is important that the manuscript justifies how it addresses this issue.
Ríos-Martín, M. Á., Folgado-Fernández, J. A., Palos-Sanchez, P. R., & Castejon-Jimenez, P. (2019). The impact of the environmental quality of online feedback and satisfaction when exploring the critical factors for luxury hotels. Sustainability, 12(1), 299.
Some suggestions are as follows:
Please use different terms in the "Title" and the "Keywords".
=> We amended terms of title and keywords.
Please use the full term when use for first time an abbreviation.
=> We reviewed abbreviation to full term properly.
The abstract should state briefly the purpose of the research, the principal results and major conclusions. An abstract is often presented separately from the article, so it must be able to stand alone.
The Introduction should end with a paragraph devoted to explaining the structure of the manuscript to the reader.
Authors should justify why they have chosen confirmatory factor analysis (CFA), based on previous work. Some are provided above. They should also indicate if there are items that were eliminated, giving reasons why.
=> We updated the specific threshold from reference that is eliminated item from confirmatory factor analysis.
Authors should explain what type of sampling they used, giving a rationale as to why it was chosen.
What software was used to do confirmatory factor analysis (CFA) and SEM. In the manuscript they state: "The data were analyzed by SPSS 24.0 and ANOVA", but ANOVA is not a software, but a statistical technique. Please clarify what ANOVA was used for.
=> We updated the threshold from adequate reference to explain elimination of the item from 1st CFA process.
It is suggested to compare the results of the present research with some similar studies which is done before.
What role can CRM systems play to refresh strategy of relationship marketing perspective and
in achieving the objectives this manuscript concludes on?
=> For the conclusion, we added CRM perspective for managerial implications from below reference, this will be valuable contribution as practical view of point.
Saura, J. R., Palos-Sanchez, P., & Blanco-González, A. (2019). The importance of information service offerings of collaborative CRMs on decision-making in B2B marketing. Journal of Business & Industrial Marketing.
It is suggested to organize Conclusion section much better. This section should present in one 250-300 words paragraph.
=> We updated conclusion adding CRM perspective.
Round 2
Reviewer 2 Report
In general, after reading the updated manuscript, the authors of this article have presented an updated version of the article, however the most of the comments reported in the first review have not been considered

Author Response
Dear Reviewer,
Thank you for your insightful feedback. We did our best to address them as below:
The authors of this article have presented an updated version of the article, where the most of the comments reported in the first review have not been considered.
Please, find below some comments and suggestion to be considered by the authors:
1.- All the authors referenced in the last section of the article must be listed in the same order as they appear in the original document. Please star in line 35 with ref. [1] and continue sequentially. It is very difficult to follow the references referenced in an unsorted list. à still missing. The first reference starts with 36. It must start with 1
-> We updated again from format guide line.
2.- All the references included in the last part are not following the rules defined in the Style Guide for MDPI Journals, please follow the main rule: “Title of the article. Journal Abbreviation Year, Volume”. This must be amended. à still missing. Journal abbreviation must be applied
-> We revised the journal name as abbreviations in the references section.
3.- In line 39 is mentioned Korea market, however the ref. 31 is referred to a Bulgaria case. This is not coherent. à still missing: Korea Tourism Org is not mentioned in the new ref 36
->Market situation changed similarly with Korean and Bulgria, local inbound business is driving due to limitation of outbound travel. We adjusted accordingly.
4.- line 46: Are the Hilton, Marriott, Hyatt names protected or registered brands? If yes, please, add “®”. Please, apply this concept to all the manuscript. à still missing: I think the brand names are not protected names, so it is necessary to mark with ®.
-> We updated ® for hotel brands.
5.- The introduction section includes several references to loyalties program without any order or key parameters. I strongly recommend to identified these key parameters and shown them in a summary table. à still missing: There is not any summary table, due to the academicals references are not sorted (bullet#1) this increase the difficulty
-> The references are now sorted updated. We hope that the reviewer understands that we did not include a summary table as it would be a simple repetition of the references in the Introduction section.
7.- The section 2.2 does not clearly demonstrate the negative effect of pandemic in the monetary sensitive. It means that I miss some bar diagrams with figures to show this impact. à still missing: Changes included in the updated revision does not improve significantly to demonstrate the negative effect of the pandemic. Changes are affecting only to a few words. It must be re-written
-> We rewrote the section as reviewer suggested. However, we hope that the reviewer understands that we did not include bar diagrams as the results are already shown in the tables in standard presentation format for SEM results.
8.- The authors should clearly indicate if tables, diagrams and figures are “source own elaboration” or not. Please review all figures and tables. And also, the tables are not following the template of the Journal. Please review the guideline information for authors. à still missing, I still don't know who made the figures or what sources they are from.
-> We updated the sources for the tables and figure.
9.- I cannot check if the survey used in this research are properly made to focus on the main hypotheses formulated in the article. In another word, I recommend to include in an annex the questionnaire to confirm that the attitudinal loyalty, behavioral loyalty, and composite loyalty are properly analyzed. à still missing: changes introduced in the Conceptual Framework part affects only a few words -> The survey is appendicized.
Once again, we really appreciate your helpful comments.
Reviewer 3 Report
ok
Author Response
Dear Reviewer,
Thank you for your feedback on our manuscript.
Sincerely,